# Recent Advances in Electrochemical Sensors and Biosensors for Detecting Bisphenol A

**DOI:** 10.3390/s20123364

**Published:** 2020-06-13

**Authors:** Somayeh Tajik, Hadi Beitollahi, Fariba Garkani Nejad, Kaiqiang Zhang, Quyet Van Le, Ho Won Jang, Soo Young Kim, Mohammadreza Shokouhimehr

**Affiliations:** 1Research Center for Tropical and Infectious Diseases, Kerman University of Medical Sciences, Kerman 7616913555, Iran; tajik_s1365@yahoo.com; 2Environment Department, Institute of Science and High Technology and Environmental Sciences, Graduate University of Advanced Technology, Kerman 76315117, Iran; f.garkani95@gmail.com; 3Jiangsu Key Laboratory of Advanced Organic Materials, Key Laboratory of Mesoscopic Chemistry of MOE, School of Chemistry and Chemical Engineering, Nanjing University, Nanjing 210023, China; kaiqiangzhang126@126.com; 4Institute of Research and Development, Duy Tan University, Da Nang 550000, Vietnam; 5Research Institute of Advanced Materials, Department of Materials Science and Engineering, Seoul National University, Seoul 08826, Korea; hwjang@snu.ac.kr; 6Department of Materials Science and Engineering, Korea University, 145, Anam-roSeongbuk-gu, Seoul 02841, Korea

**Keywords:** electrochemical sensor, biosensor, bisphenol A, electrodes

## Abstract

In recent years, several studies have focused on environmental pollutants. Bisphenol A (BPA) is one prominent industrial raw material, and its extensive utilization and release into the environment constitute an environmental hazard. BPA is considered as to be an endocrine disruptor which mimics hormones, and has a direct relationship to the development and growth of animal and human reproductive systems. Moreover, intensive exposure to the compound is related to prostate and breast cancer, infertility, obesity, and diabetes. Hence, accurate and reliable determination techniques are crucial for preventing human exposure to BPA. Experts in the field have published general electrochemical procedures for detecting BPA. The present timely review critically evaluates diverse chemically modified electrodes using various substances that have been reported in numerous studies in the recent decade for use in electrochemical sensors and biosensors to detect BPA. Additionally, the essential contributions of these substances for the design of electrochemical sensors are presented. It has been predicted that chemically modified electrode-based sensing systems will be possible options for the monitoring of detrimental pollutants.

## 1. Introduction

It is widely accepted that bisphenol A(BPA; [2,2-bis(4-hydroxyphenyl)propane]) (Figure 1) is a major monomer used for synthesizing polycarbonate and epoxy resins. Moreover, it has been used to synthesize polysulfone resin, unsaturated polyester resin, and polyphenylene oxide resin. It has been considered as a major chemical with high output in response to large demand. Diverse food storage or packaging substances are generated from polycarbonate, like feeding and water bottles, beverage containers, tableware, and microwave ovenware [1]. Epoxy resins are also used for metal coatings, including bottle tops, water supply pipes, and food cans. Because of thermal treatment during the processing phase, BPA leaches into the food products from the packaging materials, causing issues for food safety. Moreover, it is also utilized in food contact materials and in other contexts such as in the production of currency notes, thermal printing papers, compact disks, powder paints, and adhesives. In addition, the material is able to accumulate in the environment owing to its unfavorable biodegradability [2].

On the other hand, endocrine-disrupting compounds (EDCs), which represent a type of exogenous disruptor compound, are able to imitate the biological activities of natural hormones, occupying hormone receptors, interfering with the transportation and metabolic procedures of natural hormones, and threatening the wellbeing of humans and animals [3]. Thus, in recent years, there has been a growing awareness of the possible adverse effects on humans and animals resulting from exposure to EDCs that can interfere with the endocrine. These disruptions cause developmental malformations, interference with reproduction, increased cancer risk, and disturbances in immune and nervous system function. Any system in the body which is controlled by hormones can be affected by EDCs [4,5]. Toxicology studies have shown that BPA is a typical EDC which can mimic the action of the hormone estrogen [6]. Estrogen is essential for female reproduction. The estrogenic activity of the BPA is due to their structural resemblances. It has been proven that BPA can cause a range of adverse health effects including impaired brain development and sexual differentiation, enhanced rates of cancer, decreased sperm quality in humans, recurrent miscarriages, reduced immune function, and reproductive system damage [7]. Therefore, for preventing noxious impacts of BPA, it is necessary to provide an efficacious monitoring system for efficient detection.

Notably, multiple procedures have been utilized for determining the contents of BPA in diverse matrices, including gas chromatography coupled with mass spectrometry [8], high-performance liquid chromatography (HPLC) [9], fluorescence [10], electrochemical techniques [11,12], and chemiluminescence [13], which are methods which have the characteristics of excellent sensitivity, proper selectivity, and repeatability. However, these methods suffer from some disadvantages such as high cost, time-consuming operation, unsuitability for on-site analysis, and the need for qualified operators and complicated instrumentations. Therefore, it is worth developing an alternative detection method for detecting trace levels of BPA.

In fact, during the past 100 years, electroanalytical chemistry and oxidation–reduction reactions as well as other charge transfer phenomena have been utilized in their respective areas. Electroanalytical chemistry is considered a main subdiscipline of analytical chemistry. With regard to the electrical signal measured, researchers classify electrochemical procedures into conductometric, potentiometric, impedimetric, voltammetric, and amperometric techniques [14,15,16,17,18]. In recent times, voltammetric and amperometric procedures have played important roles in the detection of BPA.

Amperometry is one of a family of electrochemical methods in which the potential applied to a sensing electrode is controlled instrumentally; the recorded current occurs as a consequence of oxidation/reduction at the electrode surface. In its simplest form, the applied potential is stepped and then held at a constant value; the resulting current is measured as a function of time. Voltammetric techniques employ a measuring current at various potential points, in contrast to the fixed potential point in the amperometric technique. Over the last decades, voltammetric procedures like cyclic voltammetry (CV), square-wave voltammetry (SWV), differential pulse voltammetry (DPV), linear sweep voltammetry (LSV), and so forth have been regarded as well-known methods to determine trace concentrations of significant environmental contaminants. It is notable that electroanalytical measurements can be readily chosen frequently in a dramatically reproducible manner to resolve numerous issues of basic significance with increased degrees of accuracy, selectivity, and sensitivity [19,20,21,22,23,24,25,26].

Electrochemical detection of BPA is based on the well-known electroactivity of the phenolic groups present in the molecule (Figure 1). The electrooxidation process of BPA involves two electrons and two protons, where the reaction mechanism is based on the oxidation of the phenolic groups from the BPA structure (Figure 2) [27].

Additionally, the oxidation products of BPA cause the fouling of the electrode surface, which is a major problem occurring during the electrooxidation of phenols due to the electropolymerization of phenolic compounds [28]. The formation of polymeric product after BPA oxidation blocks the active surface area of the electrodes and delays important electrodes processes [29].

In order to avoid and reduce this problem, electrode surface modification has been widely used to inhibit the electrode surface fouling phenomena. Electrode modification has been viewed as a process where a bare electrode surface is covered with a biological or chemical compound known as a modifier, or the compound enters the electrode matrix, thereby altering its electrochemical features. According to studies, electrode surfaces could be modified by the inorganic, organic, and organic–metal electroactive compounds, biomaterials, polymer films, nanocomposites, metals, and metal oxide nanoparticles (NPs) [30,31,32,33,34,35,36]. Finally, greater sensitivity, stability, and selectivity toward analytes and better electrocatalytic activities are obtained for modified electrodes because of the utilization of substances with larger surface areas that stop unfavorable reactions, enhance the electron transfer rates, and augment the currents, thereby reducing the overpotential.

Researchers have utilized diverse electrodes and electrochemical procedures for detecting BPA. Therefore, the present research aims to examine the function of the modified electrodes applied for the identification and quantification of BPA.

## 2. Electrochemical Sensing and Biosensing of BPA

Recently, researchers have emphasized the utilization of the carbon nanomaterials and noble metals as well as metal oxide NPs, conducting polymers, ionic liquids, molecularly imprinted polymers, and biomolecules in electrochemical sensors for selectively and sensitively detecting BPA at reduced concentrations.

### 2.1. Carbon Nanomaterial-Based Electrochemical Sensors

Recently, nanomaterials have been largely considered in electrocatalysis. The typical dimension of nanomaterials is considered to be around 1–100 nm. Moreover, the versatility of the carbon atom is due to the variations in its chemical bonds, ranging between sp^3^, sp^2^, and sp^1^, and their combinations for yielding crystalline or amorphous solids. Over the past 20 years, many researchers have shifted their attentions to the electrochemical utilization of carbon nanomaterials (CNMs). In addition, such substances can be categorized into zero-dimensional NPs like quantum dots and fullerenes, one-dimensional (1D) structures such as carbon nanotubes (CNTs) and carbon nanofibers, two-dimensional (2D) layered substances like graphene (Gr), and three-dimensional (3D) structures like Gr–CNT foams and hybrids. Furthermore, the electrochemical activity of various allotropes has a strong dependence on the hybridization structure and state. In comparison to other nanomaterials, CNMs exhibit desirable characteristics like larger surface area/volume ratios, higher chemical stability, lower costs, wider potential windows, relatively inert electrochemistry, and richer surface chemistry for diverse redox reactions [37].

#### 2.1.1. CNT-Based Electrochemical Sensors

CNTs are considered to be good materials because of their high surface areas, high conductivity, and simplified functionalization, which could additionally enhance the function of electrochemical analyses. A key feature of CNTs in electrochemical sensors is their fast response with a lower limit of detection (LOD) which is caused by their higher surface area, lower overvoltage, and quick electrode kinetics. It is notable that using CNTs provides the benefit of a more rapid electron transfer rate in comparison to bulk carbon electrodes [38].

In their study, Li et al. designed a strongly sensitive technique to detect BPA with the use of a glassy carbon electrode (GCE) modified with carboxylated multiwalled CNTs (c-MWCNTs/GCE) [39]. The carboxylated groups grafted on the MWCNT surface remarkably enhanced the oxidation current of c-MWCNTs/GCE in comparison to that of the MWCNTs/GCE and bare GCE. Then, a sensitive oxidation peak was observed at 550 mV in linear sweep voltammograms at a pH equal to 7. Trace BPA levels could be determined in a linear concentration ranging from 10 to 104 nM, with a correlation coefficient equal to 0.9983 and an LOD (signal/noise (S/N) = 3) equal to 5.0 nM. Finally, their technique was shown to successfully detect BPA in food packaging.

In addition, Gao et al. provided a strongly sensitive electrochemical sensor to detect BPA in aqueous solution with a single-walled CNT (SWCNT)/β-cyclodextrin (β-CD) conjugate (SWCNT/β-CD-modified GCE) [40]. The good electrocatalytic features of SWCNTs and the inclusion capability of β-CD to BPA were demonstrated. SWCNT/β-CD-modified GCE displayed greater function for BPA oxidation, improving its oxidation peak current, lowering its oxidation potential, and lowering LOD for BPA(1.0 nM at an S/N ratio of 3). Finally, the sensor was utilized for identifying BPA leached from real plastic samples with acceptable recovery of 95%–103%.

Moreover, Cosio et al. dealt with the fabrication of an electrochemical sensor to detect BPA by means of a GCE modified with hydroxyl group (MWCNT-OH)-functionalized MWCNTs [41]. According to the analyses, the flow injection amperometric analysis system of BPA linearly responded in a concentration range of 1 to 24 µg L^−1^, with the LOD equal to 0.81 µg L^−1^. Then, the current showed a steady-state value with a very quick response time of <10 s. Consequently, their technique was utilized to identify BPA in real samples (water contained in the plastic and baby bottles) with reasonable outputs in accordance with the results observed in an independent HPLC procedure.

#### 2.1.2. Gr-Based Electrochemical Sensors

According to studies, Gr is considered as a fundamental structure in graphitic material and as aone-atom-thick planar sheet of sp^2^ bonded carbon atoms in a honeycomb crystal lattice. Since the physical exfoliation by Novoselov which led to the synthesis of Gr in 2004, Gr has been widely used to modify the electrode owing to its increased electrical conductivity, lower costs, larger surface area, and higher tensile strength [42].

In their research, Ntsendwana et al. utilized Gr for electrochemical identification of BPA [43]. Gr sheets were generated through the soft chemistry route, participating the graphite oxidation and chemical reduction. Then, X-ray diffraction (XRD) and Raman spectroscopy as well as Fourier transform infra-red (FT-IR) were utilized to feature the as-synthesized Gr. According to the analyses, Gr had an amorphous structure in comparison to the pristine graphite according to the XRD spectra. Based on FT-IR, graphene exhibited COOH and OH groups because of incomplete reduction. According to Raman spectroscopy, multi-layered Gr was formed as a result of the lower intensity of the 2D-peak. Thus, Gr with a convenient drop-and-dry procedure was utilized to modify the GCE. Consequently, CV was applied for studying the electrochemical features of the procured Gr-modified GCE with potassium ferricyanide as the redox probe. In addition, the provided Gr-modified GCE showed better electron kinetics and a current equal to ~75% as compared to the unmodified GCE. Finally, this modified electrode was employed to detect BPA. Based on the optimal condition, the BPA oxidation peak current experienced a linear variation, with concentrations ranging from 5 × 10^−8^ to 1 × 10^−6^ M, with an LOD equal to 4.689 × 10^−8^ M. Their procedure has thus been utilized for detecting BPA in plastic mineral water bottles.

#### 2.1.3. Fullerene-Based Electrochemical Sensors

According to research in the field, fullerene was rapidly investigated for utilization in electrocatalysis. In fact, researchers utilized fullerene to modify electrode surfaces, as it has chemical stability, high purity, is easy to implement, and has a reproducible electrocatalytic response [44].

In their study, Rather et al. presented a precise, selective, and sensitive electrochemical sensor to detect BPA using green technology [45]. Therefore, a fullerene-constructed electrochemical sensor was designed for the ultrasensitive detection of BPA. Then, electrochemical impedance spectroscopy (EIS), chronocoulometry, and scanning electron microscopy (SEM) were used to characterize the homemade sensor. Afterwards, researchers assessed the impacts of measuring variables such as loading of fullerene and pH on the sensor analytical functions. According to their analyses, their sensor exhibited very good electrocatalytic activities, reducing anodic overpotential and causing notable augmentation of the BPA anodic current in comparison to the electrochemical function based on GCE. In addition, Rather et al. computed diverse kinetic variables such as the electron transfer number (n), electrode surface area (A), diffusion coefficient (D), and charge transfer coefficient (α). Based on the optimized condition, the oxidation peak current had a linear relationship in a concentration range of 74 nM to 0.23 µM, with a LOD equal to 3.7 nM. Consequently, their sensor was utilized for detecting BPA in wastewater samples with encouraging analytical utilization to directly identify BPA at trace levels.

### 2.2. Noble Metal NP-Based Electrochemical Sensors

It is of note that noble metal NPs have been considered by researchers for BPA detection due to their good electrical conductivity, higher surface areas, and greater catalytic activity. Such metal NPs would be beneficial to reduce the overpotentials of the electroanalytical reactions and maintain redox reaction reversibility. In addition, noble NPs could be readily electrodeposited on the working electrode surface to enhance overall surface characteristics [46]. Specifically, gold NPs (AuNPs) show the mentioned features and are biologically compatible. Thus, the Au NP-modified electrodes would be a promising method for the detection of BPA.

In another study, Chen et al. built 3D gold (111) facet-oriented nano-dendrites on a GCE by one-step electrodeposition of AuCl_4_ˉin the presence of L-asparagine [47]. Researchers functionalized the gold nano-dendrites with 4-mercaptobenzoic acid (4-MBA), finding greater catalytic performances for sensitively and selectively detecting BPA with DPV. However, BPA oxidation peak currents at 514 mV linearly responded to BPA concentrations in the range of 0.05 to 55.0 μM (R^2^ = 0.995), with an LOD equal to 1.2 nM (S/N = 3). Finally, their modified electrode was used for the determination of trace amounts of BPA in (spiked) real samples, with reasonable outputs.

### 2.3. Metal Oxide NP-Based Electrochemical Sensors

Strongly sensitive electroanalytical instruments using nanostructured metal oxides are considered to be affordable and have improved selectivity in the case of coupling to the biorecognition molecules. Therefore, versatile procedures have been used to create metal oxide NPs (MO NPs) with diverse morphologies. Such MO NPs exhibit diverse kinds of electrical features because of their sizes, stability, and higher surface areas. In fact, key functions of the metal NPs in electroanalysis include a toughened conductive sensing interface, with the catalytic features of NPs enabling their expansion with metals and the electrical contact of redox centers with the transducer surface [48]. In addition, magnetite (Fe_3_O_4_) NPs are considered a novelty in electrochemical sensing. In fact, the increased electrical conductivity of Fe_3_O_4_ at the room temperature is caused by the electron-hopping procedures between Fe^2+^ and Fe^3+^ ions. Moreover, iron oxide NPs have been intensively utilized for modifying the electrodes to determine multiple analytes.

In their study, Mohammadzadeh Jahani et al. presented the electrochemical activation of 2-(4-((3-(trimethoxysilyl)propylthio)methyl)1-H1,2,3-triazol-1-yl) acetic acid–Fe_3_O_4_ NP (Fe_3_O_4_ NP-derivative)-modified screen printed electrodes (SPEs), referred to as Fe_3_O_4_ NP-derivative/SPEs, for the electroanalytical detection of BPA [49]. As compared to the bare electrode, the Fe_3_O_4_ NP-modified electrode notably enhanced the oxidation peak current of BPA and reduced the oxidation overpotential, suggesting a great enhancement in the detection sensitivity of BPA. In fact, the BPA oxidation peak current obtained via the DPV depicted a linear enhancement with a concentration in a range of 0.03 to 700.0 μM when the Pearson correlation coefficient was equal to 0.9998 and the LOD was equal to 0.01 μM.

### 2.4. Polymer-Based Electrochemical Sensors

Despite the use of traditional organic polymers, these sensors display specific features such as electrical conductivity, higher electron affinity, and redox activities. In fact, conducting polymers have been regarded as promising electrocatalytic substances exhibiting significant benefits for the design of electrochemical sensing instrumentations. Moreover, solvent evaporation, dip coating, electrochemical polymerization, radio frequency plasma discharge, electrochemical deposition, cross linking, and spin coating are traditional techniques used to modify electrode surfaces by coating a thin conducting polymer film. In addition, poly(3,4-ethylenedioxythiophene) (PEDOT) is regarded as a significant material because of its regular and organized chemical structure, with very good stability and conductivity [50].

In their study, Mazzotta et al. examined BPA electrochemical behaviors over PEDOT-modified GCEs via CV curves [51]. BPA oxidation on the PEDOT film generated a BPA polymer (pBPA) with good redox activities, with cathodic and anodic peaks at 0.01 and 0.15 V, respectively. Therefore, they approximated the content of the deposited pBPA by electrochemical and spectroscopic analyses via X-ray photoelectron spectroscopy (XPS). Then, Mazzotta et al. determined effects of the scan rate and pH on pBPA film oxidation behaviors. According to the analyses, oxidation current experienceda linear variation with BPA concentration in a range of 90 to 410 μM, with a LOD equal to 55 μM. Consequently, outputs of the amperometric BPA determination were gathered with a repetitive potential step program to give a linear response to BPA in a concentration range of 40 to 410 μM, with an LOD equal to 22 μM and sensitivity equal to 1.57 μAμM^−1^ cm^−2^. The sensor exhibited acceptable features of reproducibility and anti-interference, showing a successful application for the detection of BPA in mineral water samples.

### 2.5. Nanocomposite-Based Electrochemical Sensors

Nanomaterials have been widely used to develop electrochemical sensors and biosensors [52]. Nonetheless, a single nanomaterial cannot satisfy the needs of electrochemical determination; there are particular shortcomings such as dispersion difficulties and the simple reunion of the metal NPs in the polymer, adhesion of the CNTs, curling and stacking of Gr lamellae, and quantum dot agglomeration. However, experts in the field functionalized or combined nanomaterials with a number of organic/inorganic functional substances to forma nanocomposite for functional utilization. Thus, for producing a novel high-level functional substance, a nanocomposite consisting of at least one sort of nanomaterial included both the advantages of the single component material and the cooperative impact of the nanostructures. Finally, the electrochemical sensor function was enhanced via various kinds of nanocomposites with diverse dimensions and structures and specific chemical, electronic, and physical features.

It is notable that combining NPs with various substances like Gr, CNT, and polymers has provided strongly capable nanocomposite sensors for detecting BPA. Moreover, specifically metallic NPs contribute significantly to the increased selectivity and sensitivity of the sensors owing to their desirable catalytic activities. Such nanocomposites increase electron transfer, surface areas, and electrical conductivity, leading to greater selectivity and sensitivity and lower LODs.

Furthermore, Fan et al. provided a selective and sensitive procedure for detecting BPA. Nitrogen-doped graphene sheets (N-GrS) and chitosan (CS) were utilized to preparea electrochemical BPA sensor [53]. In comparison to Gr, N-GrS showed satisfactory electron transfer capability and electrocatalytic properties, enhancing the response signal toward BPA. In addition, CS exhibited very good film-forming capability and improved electrochemical behaviors of the N-GS-modified electrode. Such a situation could be attributed to the high catalytic activity, conductivity, and absorption capacity of the N-GrS. This sensor sensitively responded to BPA in a range of 1.0 × 10^−8^ to 1.3 × 10^−6^ M, with a LOD equal to 5.0 × 10^−9^ M based on the optimum condition. Ultimately, their sensor was utilized for BPA determination with reasonable outputs in the water samples.

In addition, Yaman et al. presented an electrochemical sensor using a AuNP/polyvinyl pyrrolidone (PVP)-modified pencil graphite electrode (PGE) for ultrasensitive detection of BPA [54]. Therefore, AuNPs were electrodeposited by constant potential electrolysis, and PVP was bound by passive adsorption on the electrode surface. Then, EIS and SEM were used to characterize the surface of the electrode. The researchers also investigated and optimized variables influencing experimental conditions. The AuNP/PVP/PGE sensor presented higher selectivity and sensitivity to detect BPA by using the square wave adsorptive stripping voltammetry (SWAdSV). Based on the optimum condition, the LOD was equal to 1.0 nM. Moreover, catalytic effects of the AuNP/PVP surface on BPA oxidation could be attributed to the greater surface areas and facilitated electron transfer with AuNP as well as stronger adsorption capability of BPA by using PVP. Finally, their sensor system was investigated to determine the quantities of BPA in the bottled drinking water, showing higher reliability.

In their study, Wan et al. created a simplified and environmental friendly electrochemical sensor on the basis of MWCNT/polythiophene/Pt nanocomposite-modified GCE (MWCNT/PTh/Pt/GCE) in order to detect BPA [55]. The schematic diagrams of the preparation of the MWCNT-PTh-Pt electrochemical sensor is shown in Figure 3. Their MWCNT/PTh/Pt electrochemical sensor for detecting BPA showed advantages such as larger specific surface areas of Pt and PTh nanostructures, good absorption features, and good electrochemical characteristics of MWCNTs. Based on the optimum condition, the as-prepared sensor linearly responded to BPA in a concentration range of 5.0 × 10^−8^ to 4.0 × 10^−7^ M, with a LOD equal to 3.0 × 10^−9^ M based on the 3σ rule. According to the outputs, their sensor showed higher selectivity and sensitivity and very good electrochemical function for BPA, representing its value in detecting BPA in the water samples.

Moreover, Messaoud et al. presented an electrochemical sensor for BPA consisting of a MWCNT and AuNP composite-modified GCE [56]. DPV and EIS were conducted. Then, the modified electrode architecture with various MWCNT loadings and distinct numbers of the deposited AuNP layers and pH impacts were demonstrated. Based on the optimum experimental condition, this sensor linearly detected BPA in a range of 0.01 to 0.7 µM, with the LOD equal to 4 nM. Thus, the selectivity of the sensor based on the traditional interferents was confirmed and its modified electrode was utilized to detect BPA in water.

In another study, Li et al. provided a simplified electrochemical sensor on the basis of the reduced graphene oxide–silver/poly-L-lysine nanocomposite (rGO–Ag/PLL)-modified GCE to detect BPA [57]. According to their analyses, the BPA response current notably increased following the modification of rGO–Ag/PLL caused by electroactive surfaces of the modified composites. Therefore, DPV was used as one of the analytical methods to quantitatively detect BPA. Next, the new electrochemical sensor linearly responded to BPA in a range of 1to80 μM, with an LOD equal to 0.54 μM. Consequently, the designed rGO–Ag/PLL/GCE sensor was utilized for detecting BPA with suitable outputs in drinking water.

Another study conducted by Hao et al. reported on an AuNPs-modified GCE. The AuNPs were loaded onto the rGO–MWCNT composite film [58]. Therefore, the rGO–MWCNT composite was procured by in-situ chemical reduction with hydrazine as a reducing agent. The AuNPs were also deposited on the rGO–MWCNT surface using a simplified CV. Then, their modified electrode was demonstrated via the SEM, energy-dispersive X-ray spectroscopy (EDS), and electrochemical techniques. In addition, it was used to analyze the electrochemical behavior of BPA. Based on the synergistic action of MWCNTs and rGO, the rGO–MWCNT composite showed reasonable dispersion, larger particular surface areas, and greater active sites. Moreover, it showed feasibility as an acceptable carrier for obtaining AuNPs with higher electrochemical activities. Outputs indicated greater electrochemical activity by the modified electrode in oxidizing the BPA. Moreover, in 0.1 M phosphate buffer (PBS, pH = 7.0), BPA determination with DPV linearly ranged between 5.0 × 10^−9^ to 1.0 × 10^−7^ M and 1.0 × 10^−7^−2.0 × 10^−5^ M. Furthermore, the LODwasequal 1.0 × 10^−9^ M (S/N *=* 3). Finally, the as-prepared modified electrode was employed for detecting BPA in river water and shopping receipt samples, revealing recovery in ranges of 97%−110% and 98%−104%, respectively.

### 2.6. Ionic Liquid (IL)-Based Electrochemical Sensors

It is widely accepted that ionic liquids are compounds containing ions which are liquid at a temperature, holding an organic cation and an organic or inorganic anion. However, room-temperature ILs (RTILs) have recently been considered in electrochemical sensors and analytical chemistry. Because of their specific features like wider electrochemical windows, insignificant vapor pressure, and higher ionic conductivity, as well as acceptable solubility, experts in the field utilized RTIL as the electrode modifier. In fact, the electrochemical sensor function was enhanced by combining the ILs with NPs [59].

In their study, Chen et al. applied 3-butyl-1-[3-(N-pyrrolyl)propyl]imidazolium as a cation, carboxylic acid-functionalized SWCNT-COO as an anion, and created a SWCNT-COO–IL nanocomposite [60]. According to the results, the as-prepared SWCNTs-COO–IL nanocomposite was confirmed by transmission electron microscopy (TEM), XPS, FTIR, Raman spectroscopy, and UV–Vis. Moreover, the SWCNT-COO–IL nanocomposite was coated on a GCE surface accompanied by cyclic voltammetric scanning for fabricating a SWCNT/poly[3-butyl-1-[3-(N-pyrrolyl)propyl] imidazolium ionic liquid} composite film-modified electrode (SWCNT/Poly-IL/GCE). Then, researchers applied the SEM and EIS to characterize the SWCNT/Poly-IL/GCE. In fact, they fullyexamined BPA electrochemical behavior with the SWCNT/Poly-IL/GCE. Based on the findings, a clear oxidation peak was observed, with a reduction peak in the reverse scanning reflecting an irreversible electrochemical procedure. Moreover, BPA oxidation peak currents exhibited a linear correlation with the scan rates in a range of 20–300 mV s^−1^, representing an absorption-controlled procedure instead of a diffusion-controlled one. In addition, DPV was utilized for BPA voltammetric sensing. Chen et al. also addressed the optimization of experimental conditions such as the film thickness, pH-value, and accumulation time, and potential, which influenced the analytical functions of the SWCNT/Poly-IL/GCE. Based on the optimum condition, oxidation peak currents depicted a linear association with BPA concentrations in a range of 5.0 × 10^−9^ to 3.0 × 10^−5^ M, with a LOD equal to 1.0 × 10^−9^ M (S/N = 3). Finally, their technique was utilized to detect BPA leaching from plastic drinking bottles.

Another investigation by Najafi et al. dealt with BPA electrochemistry by voltammetric procedures at the surface of the CPE modified by a ZnO/CNT nanocomposite and room temperature IL of 1,3-dipropylimidazolium bromide [61]. Researchers carefully monitored ZnO/CNTs and the IL ratio on the electrode surface due to the charging current. Complete separation of anodic peaks of BPA and Sudan I in their mixture was provided. In fact, at pH = 7.0, two peaks were separated, at ca. 0.47 and 0.70 V. Thus, it was possible to detect BPA in the presence of Sudan I with more than 8.7 times the current excess of BPA. It is notable that researchers observed a linear increase of the peak currents of SWV of BPA and Sudan I, with concentrations ranging from 0.002–700 µM BPA to 0.2–800 µM Sudan I. Analyses showed 9.0 nM of LOD for BPA and 80 nM of LOD for Sudan I. Finally, the modified electrode was used to assay BPA in the food samples. This research provided a simplified method for selective determination of BPA in the presence of Sudan I.

### 2.7. Molecularly Imprinted Polymer-Based Electrochemical Sensors

Molecularlyim-printed polymers (MIPs) are considered to be synthetic polymers with molecular identification capability for the target analyte [62]. According to studies, molecular imprinting reveals the molecular determination sites in a polymer via synthesis in the presence of the target template. Therefore, the supplementary interaction of the functional monomers with the template molecule will be preserved in their spatial arrangement by polymerization and additionally stabilized by cross-linking the polymer. Hence, the resultant MIP canselectively determine the target analyte in the template-derived sites. It is notable that molecular imprinting of polymers is presently the most versatile, generic, affordable, and scalable method for creating synthetic molecular receptors. Increasing attention has been paid to MIPs due totheir high selectivity and affinity, which are crucial for their use as true alternative receptors. Another advantage is their greater stability, which has been shown to be better than that of natural biomolecules. Moreover, their simplified procurement allows for the availability of a receptor for an analyte far more quickly than that in the case of dependence on the antibody production. The other benefit is template versatility, whereby the MIPs can determine analytes, which provide the challenging target antigens for antibodies. Finally, MIPs display simple adaptation for functional use in specific sensors and assays. Hence, a sensor that combines electrochemical and molecularly imprinted methods can be assumed to be able to achieve BPA determination.

Deng et al. introduced an electrochemical sensor on the basis of an acetylene black paste electrode (ABPE) modified with the molecularly imprinted CS–Gr composite film to sensitively and selectively detect BPA [63]. Researchers examined multiple prominent variables influencing the sensor function and optimized them. Analyses indicated rapid response and sensitive quantification of BPA by the imprinted sensor. Based on the optimized condition, BPA showed a linear detection range of 8.0−1.0 µM to 1.0−2.0 µM, with a LOD equal to 6.0 nM (S/N = 3). In addition, their sensor displayed very good particular determination of the template molecule in comparison to structurally similar and co-existing materials. Ultimately, their imprinted electrochemical sensor was successfully used for detecting BPA in plastic bottled drinking water and canned beverages.

In their study, Tan et al. presented an electrochemical sensor on the basis of a molecularly imprinted polypyrrole/graphene quantum dot (MIPPy/GQD) composite fordetectingBPA in the water samples [64]. They procured a MIPPy/GQD composite layer through pyrroleelectro polymerization on a GCE using BPA as a template. Results showed the specific detection of BPA in the aqueous solutions by the MIPPy/GQD composite, causing the decreased peak currents of K_3_[Fe(CN)_6_] at the MIPPy/GQD-modified electrode in DPV and CV. Moreover, a linear association was observed when BPA concentrations were in the range of 0.1 to 50 μM, and the response value in DPV with the LOD was equal to 0.04 μM (S/N = 3). Finally, their sensor was utilized to determine BPA in tap and sea water samples, with a recovery range of 94.5%–93.7%. Hence, this procedure provided a robust instrument to rapidly and sensitively detect BPA in environmental samples.

### 2.8. Biomolecule-Based Electrochemical Biosensors

The biosensor is a potential analytical device which is used as a biorecognition element (e.g., aptamers, enzymes, antibodies, etc.) to interact (i.e., recognize or bind) with an analyte. A biosensor usually consists of three elements: a biorecognition element, a biotransducer, and an electronic system that includes a processor, signal amplifier, and display [65]. The recognition part is often termed as a bioreceptor, and exploits biomolecules as receptors to interact with a specific analyte. The affinity of the biorecognition elements toward the targeted analytes provides specific detection and also increases the selectivity of the detection.

#### 2.8.1. Aptamer-Based Electrochemical Biosensors

Aptamers are considered to be artificial, short, single-stranded DNA/RNA oligonucleotides which are selected invitro through the systematic evolution of the ligands by the exponential enrichment. Aptamers can bind with different kinds of molecules with high specificity and high affinity. In comparison with antibodies, the use of aptamers offers benefits like smaller size, controllable synthesis procedure, and ease of modification by a variety of molecules [66]. These make aptamers exciting candidates in electrochemical determination. Aptamer-based biosensors also known as aptasensors.

Research conducted by Zhou et al. provided a simplified and label-free electrochemical aptasensor to detect BPA on the basis of an Au NP-dotted Gr (AuNP/Gr) nano-composite film-modified GCE [67]. Figure 4 schematically depicts the mechanism of the strategy in this work to fabricate the electrochemical aptasensor. This aptamer is directly immobilized on AuNP/Gr nanocomposite film-modified electrode for BPA capture. Then, electrochemical probe of ferricyanide is utilized for examining the interaction between BPA and the aptamer. Outputs indicate an acceptable current response to detect BPA through the resultant AuNP/Gr layer. The researchers used an atomic force microscope (AFM), SEM, and CV to characterize the strongly biocompatible and conductive nanostructure of the AuNP/Gr nanocomposite. Moreover, they found linearity of the peak current change of ferricyanide with BPA concentrations in a range of 0.01 and 10 µM, with a LOD equal to 5 nM. It should be mentioned that their aptasensor displayed rapidness, convenience, and affordability for efficient sensing of BPA. Finally, it was successfully utilized to detect BPA in milk products, and average recovery was considered to be 105%.

In another study, Deiminiat et al. produced a label-free electrochemical aptasensor for sensitively and selectively detecting BPA on the basis of a functionalized MWCNT/AuNP(f-MWCNT/AuNP) nanocomposite film-modified gold electrode [68]. Then, the f-MWCNT/AuNP nanocomposite was chemically synthesized and the structure of the procured nanocomposite was shown by UV-Vis spectrophotometry, FT-IR spectrometry, XRD, and TEM. Afterwards, the construction procedure of the electrochemical sensor was examined using the CV and EIS in the presence of [Fe(CN)_6_]^3−^/[Fe(CN)_6_]^4−^ as one of the electrochemically active probes. Next, molecular dynamic (MD) simulations were utilized to study the interactions between BPA and the respective aptamer molecules. The effects of numerous variables on the aptasensor function were examined and optimized. Based on the optimum experimental condition, SWV was utilized as a sensitive analytical method to detect BPA in the solutions, and thus an acceptable linear association was found between BPA concentration and peak current in a range of 0.1 to 10 nM, with anLOD equal to 0.05 nM. In addition, effects of the interfering samples on determining BPA were determined, resulting in the high selectivity of the introduced aptasensor toward BPA. Outputs showed the sensor’s acceptable stability and reproducibility. Ultimately, this aptasensor was utilized to determine BPA in real samples of mineral water, milk, and orange juice.

Moreover, Baghayeri et al. used an electrochemically aptamer-based method to detect BPA [69]. They used AuNPs immobilized on the conjugate between the MWCNT and thiol-functionalized magnetic nanoparticles (MWCNT/Fe_3_O_4_-SH). The designation and fabrication strategies of GCE/MWCNT/Fe_3_O_4_-SH@Au/aptamer/6-mercapto-1-hexanol (MCH) for the sensing of BPA molecules were also demonstrated. Then, FT-IR, TEM, vibrating sample magnetometry, elemental mapping analysis, field emission SEM, and energy dispersive XRD were used for characterizing nanocomposite. Notably, the aptasensor that acted at 0.20 V (vs. Ag/AgCl) possessed a linear response at a concentration ranging from 0.1 to 8 nM BPA, with a lower LOD equal to 0.03 nM and higher sensitivity (86.43 μA nM^−1^ cm^−2^). In addition, voltammetric experiments were performed with the hexacyanoferrate redox system as one of the electrochemical probes. Outputs indicated a synergistic electrochemical enhancement due to the presence of magnetic NPs, MWCNTs, and AuNPs. Therefore, this new technique is strongly sensitive, selective, effective, and environmentally friendly. It has been utilized for detecting BPA in real spiked samples.

#### 2.8.2. Enzyme-Based Electrochemical Biosensors

Enzymes are very efficient biocatalysts, which have the ability to specifically react with their substrates (target analytes) and to catalyze their transformation. Over the last decade, considerable interest has been focused on enzyme-based electrochemical biosensing systems due to their higher catalytic activities and sensitivity, simplified action, facileminiaturization, and selectivity. These systems are thus considered an alternative to conventional methods [70].

Li et al. reported on the biosensing of BPA using rGO–1,3-di(4-amino-1-pyridinium) propane tetrafluoroborate IL (rGO–DAPPT) as an interface to immobilize tyrosinase (Tyr) [71]. The Tyr-DAPPT–rGO/GCE was characterized with SEM and EIS. The biosensor showed a linear response to BPA in the concentration range of 1.0 × 10^−9^ to 3.8 × 10^−5^ M. The LOD was calculated to be 3.5 × 10^−10^ M. The biosensor was further applied to analysis of BPA leaching from plastic drinking bottles, showing satisfactory results.

Kunene et al. reported a novel electrochemical biosensor for the detection of BPA using a carbon-screen-printed electrode (CSPE) modified with MWCNTs that werefunctionalized with silver-doped zinc oxide NPs, on which the laccase enzyme was immobilized (Lac/Ag–ZnO/MWCNTs/CSPE) [72]. The presence of Lac on the surface of the modified electrode reduced the charge transfer resistance of the redox couple and thereby improved the sensitivity towards the oxidation of BPA. The biosensor displayed outstanding performance for detecting BPA with a linear range of 0.5–2.99 μM and an LOD of 6.0 nM. Furthermore, this proposed laccase biosensor was more selective and stable, with a highly reproducible response factor (relative standard deviation (RSD) of 0.86%). It was able to adequately quantify BPA in plastic bottle samples.

Liu et al. fabricated an electrochemical Tyr enzyme biosensor using a highly conductive sugarcane-derived biochar nanoparticle (BCNP) as a transducer (BCNP/Tyr/Nafion/GCE) for the sensitive detection of BPA [73]. The prepared biochar nanoparticles had large specific surface areas, good conductivity, and were biocompatible and environmentally friendly, characteristics which were beneficial for enhancing the activity of the enzyme. This sensor could detect BPA with good sensitivity of 0.985 µA µM^−1^ cm^−2^ in the linear range of 0.02 to 10 µM, and the lowest LOD of 3.18 nM. Moreover, this biosensor exhibited high reproducibility and good selectivity over other ionic interferents. The biosensor was also successfully utilized for real water detection with high accuracy, as validated by HPLC.

#### 2.8.3. Antibody-Based Electrochemical Biosensors

Antibodies are proteins produced by the immune system and contain antigenrecognition sites, which bind to their specific related antigens by noncovalent interactions with relatively high affinity [74]. The antibody/antigen interaction is the most important phenomenon in immunosensors. Immunosensors are specific, sensitive, and can offer real-time analysis of multiple targets in an automated platform.

Huang et al. reported a new and label-free electrochemical immunosensor for sensitive detection of BPA [75]. MWCNTs and AuNPs were modified on the GCE surface to enhance current response. The anti-BPA was immobilized on the modified electrode through AuNPs. After that, the possible remaining active sites on the anti-BPA/AuNP/MWCNT/GCE were blocked through incubating in bovine serum albumin (BSA). Rutin was used as the redox probe to construct electrochemical immunosensor of BPA. The peak current change due to the specific immuno-interaction between anti-BPA and BPA on the modified electrode surface was utilized to detect BPA. Under optimum conditions, the linear range of calibration curve based on the relationship between current response and BPA concentration was from 1.0 × 10^−8^ to 1.0 × 10^−6^ M, with detection limit of 8.7 × 10^−9^ M (S/N = 3). The proposed immunosensor showed good reproducibility, selectivity, and stability and was successfully applied to the determination of BPA in food fresh-keeping film.

A list of electrodes, electrochemical methods, and results for electrochemical sensors and biosensors in the detection of BPA are summarized in Table 1.

## 3. Conclusions

Experts in the field have shown great interest in the utilization of the electrochemical sensors to detect BPA. In general, an excellent electrochemical sensor would have the features of simplicity, portability, precision, sensitivity, selectivity, repeatability, acceptable mechanical and chemical stability, affordability, and shorter response and recovery time. In addition, it should simultaneously determine numerous analytes with low LODs. Modifying the electrode surface through proper modifiers causes an increased efficient surface and rate of electron transfer reactions, which finally enhances electrocatalytic features and the function of the electrode. CNMs, noble metals, metal oxide NPs, conducting polymers, nanocomposites, ionic liquids, molecularly imprinted polymers, and biomolecules have been considered amongst modifiers for successful applications in BPA determination in electrochemical sensors. Clearly, they have a considerable impact on the selectivity, sensitivity, repeatability, and accuracy of the sensors.

Although great achievements have been made in the past decades in this field, many challenges still exist for further research on electrochemical BPA sensors:

(1) The properties of nanomaterials depend on the size, shape, oxidation level, degree of purity, structural defects, and degree of dispersion, since these factors greatly affect the electrical conductivity, catalytic activity, and physical chemical binding interactions and kinetics between nanomaterials and other functional materials. In addition, some synthetic routes of nanomaterials are tedious and special conditions are required. Therefore, we need to search for a new synthesis method to produce uniform and stable nanomaterials and composites for the design of BPA sensors. 

(2) The leakage or detachment of modifiers from the sensor surface during the usage may have a profound influence on the overall performance of the sensor. That is, the leakage or detachment of modifiers may reduce the number of sites on the sensing surface and reduce the electronic conductivity of the sensor. Therefore, effective and robust immobilization methods for modifiers in BPA electrochemical sensors should be considered carefully. 

(3) Some electrode preparations are complex, time consuming, and costly. Therefore, easy and low-cost modification of the electrode surface is very important in fabricating electrochemical BPA sensors.

(4) The matrix effect is crucial when applying the sensors for real sample analysis. Samples like food and water always contain a mixture of ions, macromolecules, and other electroactive species that would induce a nonspecific signal in the BPA detection process. For this, one research direction lies in biorecognition elements which should be specific to BPA; another relates to effective sample pretreatment methods. However, the efficient immobilization of recognizing elements for transducers remains a major challenge in biosensor technology.

(5) The commercialization of BPA electrochemical sensors has been hampered due to some technical barriers. In many cases, the main limitations in commercialization of electrochemical sensors are poor analytical performance in real samples, complex detection procedures, and a lack of cost-effective production methods. Therefore, further improvements in terms of miniaturization, portability, minimizing matrix effect, ability to conduct continuous measurements, validation parameters, and competitive cost are necessary.

Future research should resolve these challenges in order to improve the electrochemical detection of BPA.

## Figures and Tables

**Figure 1 sensors-20-03364-f001:**
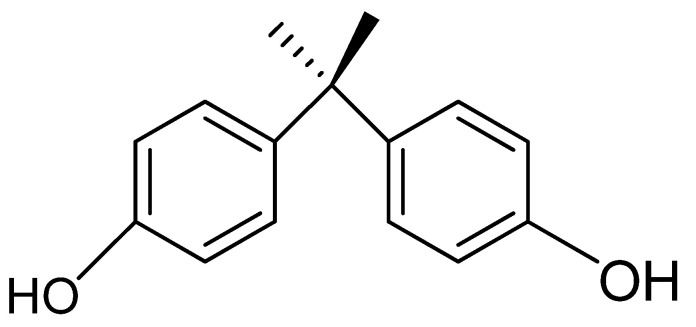
Chemical structure of bisphenol A (BPA).

**Figure 2 sensors-20-03364-f002:**
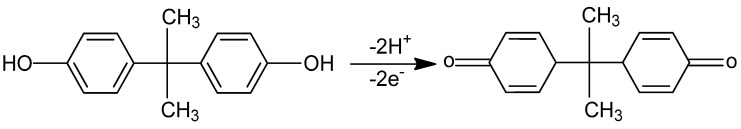
Electrooxidation mechanism of BPA.

**Figure 3 sensors-20-03364-f003:**
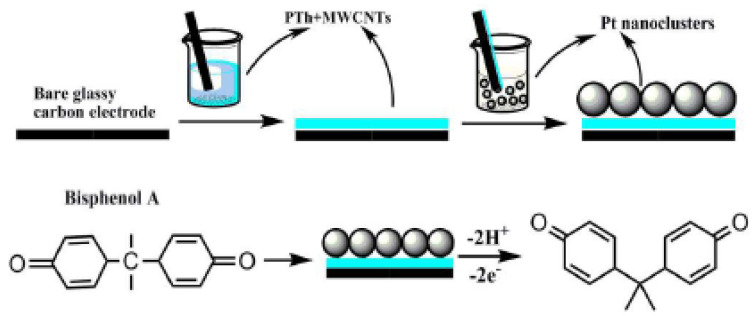
The schematic diagram of preparation of the multi-walled carbon nanotube (MWCNT)/polythiophene (PTh)/Pt electrochemical sensor. Reprinted with permission from [55].

**Figure 4 sensors-20-03364-f004:**
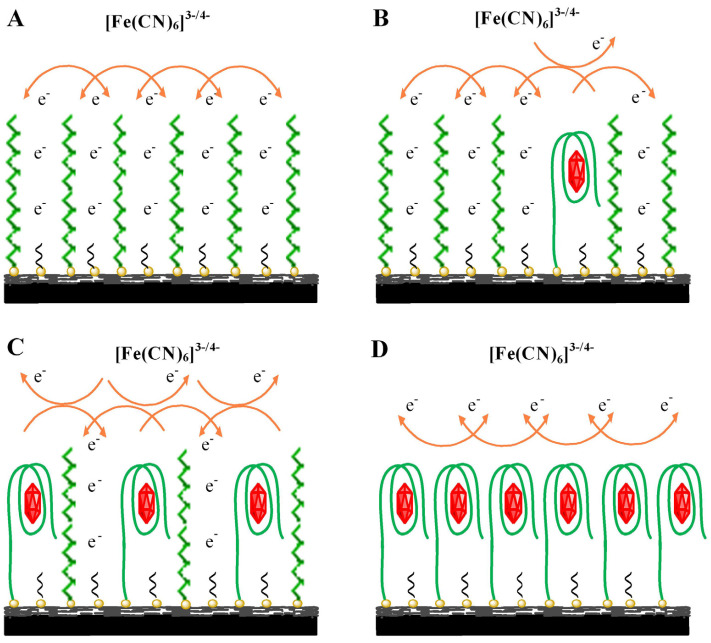
Schematic illustration of the designed strategy for the determination of BPA. The tunnel gates remain open in the absence of BPA (**A**). Addition of BPA (lower concentration) results in the closing of some of the gates due to conformational aptamer changes (**B**). An increase of BPA further increases the number of closed gates (**C**). At higher concentrations almost all the gates are closed (**D**). Reprinted with permission from [67].

**Table 1 sensors-20-03364-t001:** Comparison of the performance of electrochemical BPA sensors and biosensors.

Ref.	Linear Range	Detection Limit	Method	Electrochemical Sensor
[39]	10–104 nM	5.0 nM	Voltammetry	c-MWCNTs/GCE
[40]	10.8 nM–18.5 µM	1.0 nM	Amperometry	SWCNT–â-CD/GCE
[41]	1–24 µg L^−1^	0.81 µg L^−1^	Amperometry	MWCNTs-OH/GCE
[43]	5 × 10^−8^–1 × 10^−6^ M	4.689 × 10^−8^ M	DPV	Gr/GCE
[45]	74 nM−0.23 µM	3.7 nM	SWV	Fullerene/GCE
[47]	0.05–55.0 µM	1.2 nM	DPV	4-MBA/gold nano-dendrites/GCE
[49]	0.03–700.0 µM	0.01 µM	DPV	Fe_3_O_4_ NP-derivative/SPE
[51]	90–410 µM	55 µM	CV	PEDOT/GCE
	40–410 µM	22 µM	Amperometry	
[53]	1.0 × 10^−8^–1.3 × 10^−6^ M	5.0 × 10^−9^ M	Amperometry	CS/N-GrS/GCE
[54]	0.03–1.10 µM	1.0 nM	SWAdSV	AuNP/PVP/PGE
[55]	5.0 × 10^−8^–4.0 × 10^−7^ M	3.0 × 10^−9^ M	DPV	MWCNT/PTh/Pt/GCE
[56]	0.01–0.7 µM	4 nM	DPV	AuNP/MWCNT/GCE
[57]	1–80 µM	0.54 µM	DPV	rGO–Ag/PLL/GCE
[58]	5.0 × 10^−9^−2.0 × 10^−5^ M	1.0 × 10^−9^ M	DPV	AuNP–rGO–MWCNTs/GCE
[60]	5.0 × 10^−9^–3.0 × 10^−5^ M	1.0 × 10^−9^ M	DPV	SWCNTs/Poly-IL/GCE
[61]	0.002–700 µM	9.0 nM	SWV	ZnO/CNT/IL/CPE
[63]	8.0−2.0 µM	6.0 nM	Derivative Voltammetry	MIP/CS/Gr/ABPE
[64]	0.1–50 µM	0.04 µM	DPV	MIPPy/GQDs/GCE
[67]	0.01–10 µM	5 nM	DPV	AuNP/Gr/GCE
[68]	0.1–10 nM	0.05 nM	SWV	f-MWCNT/AuNP/Au
[69]	0.1–8 nM	0.03 nM	DPV	MWCNT/Fe_3_O_4_-SH@Au/
				aptamer/MCH/GCE
[71]	1.0 × 10^−9^–3. 8 × 10^−5^ M	3.5 × 10^−10^ M	Amperometry	Tyr-DAPPT–rGO/GCE
[72]	0.5–2.99 µM	6.0 nM	DPV	Lac/Ag–ZnO/MWCNTs/CSPE
[73]	0.02–10 µM	3.18 nM	Amperometry	BCNP/Tyr/Nafion/GCE
[75]	1.0 × 10^−8^–1. 0 × 10^−6^ M	8.7 × 10^−9^ M	LSV	BSA/Anti-BPA/AuNP/MWCNT/GCE

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
