# Peer review of "Recent Advances in Electrochemical Sensors and Biosensors for Detecting Bisphenol A"

_sensors, 2020, doi:10.3390/s20123364_

Round 1

Reviewer 1 Report

This short overview of electrochemical and biosensor-based detection methods for bisphenol A will be useful to those looking for a single source of information on the subject. It is thorough and well-illustrated although some of the English usage (‘growingly’ instead of ‘growing’, ‘nobel’ instead of ‘noble’, etc) needs attention. The extensive list of references shows that the review covers all the key developments in the last decade or so. Subject to a check of the English I am happy to support publication.

Author Response

-This short overview of electrochemical and biosensor-based detection methods for bisphenol A will be useful to those looking for a single source of information on the subject. It is thorough and well-illustrated although some of the English usage (‘growingly’ instead of ‘growing’, ‘nobel’ instead of ‘noble’, etc) needs attention. The extensive list of references shows that the review covers all the key developments in the last decade or so. Subject to a check of the English I am happy to support publication.

Answer: We corrected the growingly and nobel words. Furthermore, we double checked the English of the manuscript to improve the readability.

Reviewer 2 Report

  1. In the first figure (page 2), the authors list different materials that can be used for functionalising electrode surface. While aptamers (as biomaterial) have been included within the figure some other common materials used for detecting Bisphenol A are missing. These includes enzymes, antibodies, as well as metal complexes, etc. Is it possible to group the materials within the figure based on their properties or material characteristics?
  2. The quality of Figure 1 (page 3) should be improved – the structures appear blurry.
  3. It is difficult to understand the health implications of exposure to Bisphenol A (second paragraph, lines 51 – 61). For example, it would be helpful to provide information on what conditions can be caused when endocrine system is interfered by endocrine disrupting chemicals.
  4. The authors list a number of analytical techniques (lines 71-74) often used for the determination of Bisphenol A including chromatography, fluorescence, etc., however there is no discussion around disadvantages of these analytical methods and why there is a continuous need to develop alternative detection platforms.
  5. Line 75 – Electroanalytical techniques were mainly presented in the context of voltammetric methods. It would be helpful to demonstrate that Bispehnol A have been also successfully detected using other electroanalytical approaches including amperometry, impedance, field effect transistors etc.
  6. Line 87 – While the authors describe challenges associated with the application of unmodified electrodes for electrochemical sensing, it would be helpful to put it in context of detecting Bisphenol A. Additional discussion on properties and redox mechanism of Bisphenol A should be included. Furthermore, it is unclear what is meant by bare electrode (e.g. graphene or graphite, glassy carbon, etc.) and why these electrodes suffer from reduced electron transfer kinetics? Is it due to adsorption of oxidation products of Bisphenol A on electrode surface during redox transformations? Discussion regarding surface characteristics and their influence on the electroanalytical performance of sensors would be helpful. It would be also helpful to include details on oxidation mechanism of Bisphenol A during electroanalytical measurements.
  7. Lines 93-95 - The authors state that for the surface modified electrodes “the compound enters the electrode matrix and alters its electro chemical features”. However, it is unclear what is meant by this statement. The detection mechanism will vary between different electrode surfaces (e.g. direct binding transduced electronically, surface interactions, etc.). It may be helpful for the reader to understand the most common interaction mechanisms between electrode surface material and Bisphenol A. In addition, discussion regarding electrode modifications needed to develop biosensors, where a biomaterial (e.g. enzyme, aptamer, antibody, etc.) which functions as recognition element is typically immobilised on the electrode surface, should be included.
  8. While the authors list numerous advantages of using modified electrodes for detecting analytes (lines 98 – 102), it would be helpful to include additional discussion on some disadvantages associated with extensive electrode surface modifications such as poor batch-to-batch reproducibility, often lengthy and complex preparation steps, etc.
  9. To improve clarity for the reader, it would be helpful to include a table summarising key electrochemical sensors developed for sensing Bisphenol A with a further breakdown to highlight sensor configuration, detection limits, electroanalytical techniques used, etc.
  10. The rationale behind the inclusion of described studies is unclear. Are these key fundamental or first demonstrations of applying a specific material for developing a Bisphenol A sensors? Are these selected based on the best performance metrics including sensitivity, selectivity or detection range, etc. (to date) based on published literature? The description on how this review is organised with associated rationale should be included.
  11. It would be helpful for the reader to include at the end of each section (e.g. CNT based electrochemical sensors, ionic liquids, etc.) a comprehensible discussion on future challenges associated with adaptation of these materials for making stable, sensitive and reproducible sensors. For example, changes in surface morphology of graphene, CNTs are known to strongly influence the response of studied molecules in electroanalytical measurements. In addition, challenges associated with high manufacturing costs, limited control over surface defects during synthesis, etc. in the context of making electroanalytical sensors for Bisphenol A should be discussed.
  12. Discussion regarding current technological challenges associated with the development and commercialisation of Bisphenol A sensors should be added to the “Conclusions” section. Discussion on the solutions that can be implemented to overcome these issues should be also included. Also, it would be helpful to provide a future outlook regarding which materials or sensor configurations demonstrate the most promising characteristics for developing stable, reproducible, sensitive and selective Bisphenol A sensors.
  13. The English language throughout the manuscript must be improved.

Author Response

-In the first figure (page 2), the authors list different materials that can be used for functionalising electrode surface. While aptamers (as biomaterial) have been included within the figure some other common materials used for detecting Bisphenol A are missing. These includes enzymes, antibodies, as well as metal complexes, etc. Is it possible to group the materials within the figure based on their properties or material characteristics?

Answer: We added electrochemical biosensors based on bisphenol A detection to the manuscript using other biological molecules such as enzymes and antibodies.

2.8. Biomolecules based electrochemical biosensors

The biosensor is a potential analytical device, which is used as a biorecognition element (e.g., aptamers, enzymes, antibodies, etc.) to interacts (i.e., recognizes or binds) with an analyte. A biosensor usually consists of three elements; a biorecognition, biotransducer, and an electronic system that include a processor, signal amplifier, and display [34]. The recognition part often termed as a bioreceptor, which exploits biomolecules as receptors to interact with a specific analyte. The affinity of the biorecognition elements toward the targeted analytes provides specific detection and also increases the selectivity of the detection.

2.8.2. Enzyme based electrochemical biosensors

There is enough information that enzymes contained the rectilinear sequence of amino acid that could wrap for forming 3D structure with the catalytic activities towards certain substrates.Over the last decade, considerable interest has been focused on enzyme based electrochemical biosensing systems due to higher catalytic activities and sensitivity, simplification, simple miniaturization and selectivity as an alternative to conventional methods [37].

Li et al. reported biosensing of BPA using rGO-1,3-di(4-amino-1-pyridinium) propane tetrafluoroborate IL (rGO-DAPPT) as an interface to immobilize tyrosinase (Tyr) [38a]. The Tyr-DAPPT-rGO/GCE was characterized with SEM and EIS. The biosensor shows a linear response to BPA in the concentration range of 1.0×10-9-3.8×10-5 M. The LOD was calculated to be 3.5×10-10 M. The biosensor was further applied to analysis BPA leaching from plastic drinking bottle with satisfactory results.

Kunene et al. reported a novel electrochemical biosensor for the detection of BPA using a carbon-screen printed electrode modified with MWCNTs that are functionalized with silver doped zinc oxide NPs on which laccase enzyme was immobilized (Lac/Ag–ZnO/MWCNTs/CSPE) [38b]. The fabrication of biosensor was illustrated in Figure 6. The presence of Lac on the surface of the modified electrode reduced the charge transfer resistance of the redox couple and thereby improved the sensitivity towards the oxidation of BPA. The biosensor displayed outstanding performance for BPA with a linear range 0.5–2.99 μM and a LOD of 6.0 nM. Furthermore, this proposed lacasse biosensor is more selective and stable with a high reproducible response factor (relative standard deviation (RSD) of 0.86%), and was able to adequately quantify BPA in plastic bottle samples.

Figure 6. A schematic representation of the electrochemical biosensor for the detection of oxidation of BPA using Lac/Ag-ZnONPs/MWCNTs/CSPE. Reprinted with permission from ref. [38b]

Liu et al. fabricated an electrochemical Tyr enzyme biosensor using a highly conductive sugarcane derived biochar nanoparticle (BCNP) as a transducer (BCNPs/Tyr/Nafion/GCE) for the sensitive detection of BPA [38c]. The prepared biochar nanoparticles had large specific surface areas, good conductivity, biocompatibility and environmental friendliness which were beneficial to enhance the activity of the enzyme. This sensor could detect BPA in good sensitivity 0.985 µA µM-1 cm-2 with linear range from 0.02 to 10 µM, and a lowest LOD 3.18 nM. Moreover, this biosensor exhibited high reproducibility and good selectivity over other ionic interferents. The biosensor was also successfully utilized for real water detection with high accuracy as validated by HPLC.

2.8.3. Antibody-based electro-chemical bio-sensors

Antibodies are proteins produced by the immune system and contain antigen-recognition sites, which bind to their specific related antigens by noncovalent interactions with relatively high affinity [39].The antibody/antigen interaction is the most important phenomenon in immunosensors. Immunosensors are specific, sensitive and can offer real time analysis of multiple targets in an automated platform.

Huang et al. reported a new and label free electrochemical immunosensor for sensitive detection of BPA [40]. MWCNTs and AuNPs were modified on GCE surface to enhance current response. The Anti BPA was immobilized on the modified electrode through AuNPs. After that, the possible remaining active sites on Anti-BPA/AuNPs/MWCNTs/GCE were blocked through incubating in bovine serum albumin (BSA). Rutin was used as the redox probe to construct electrochemical immunosensor of BPA. The peak current change due to the specific immuno-interaction between anti-BPA and BPA on the modified electrode surface was utilized to detect BPA. Under optimized conditions, the linear range of calibration curve based on the relationship between current response and BPA concentration was from 1.0×10-8 - 1.0×10-6 M with detection limit of 8.7×10-9 M (S/N = 3). The proposed immunosensor showed good reproducibility, selectivity, stability and was successfully applied to the determination of BPA in food fresh-keeping film.

-The quality of Figure 1 (page 3) should be improved – the structures appear blurry.

Answer: We improve the quality of the Figure 1.

The revised Figure 1 now reads:

Figure 1. Chemical structure of BPA.

-It is difficult to understand the health implications of exposure to Bisphenol A (second paragraph, lines 51 – 61). For example, it would be helpful to provide information on what conditions can be caused when endocrine system is interfered by endocrine disrupting chemicals.

Answer: We have added explanations in this regard to the manuscript.

The revised part now reads:

Thus, in recent years, there has been a growing awareness of the possible adverse effects in humans and animals from exposure to EDCs that can interfere with the endocrine system. These disruptions cause developmental malformations, interference with reproduction, increased cancer risk, and disturbances in the immune and nervous system function and therefore, any system in the body which is controlled by hormones can be affected by EDCs [4,5]. Toxicology studies have shown that BPA is a typical EDC which can mimics the action of hormone estrogen [6]. Estrogen is essential for female reproduction. The estrogenic activity of the BPA is due to the structural resemblances with the estrogenic hormones. It is generally proved that BPA can cause a range of adverse health effects including impaired brain development, sexual differentiation, enhance rate of cancer, decrease of sperm quality in humans, recurrent miscarriages, reduced immune function and damage reproduction [7].

-The authors list a number of analytical techniques (lines 71-74) often used for the determination of Bisphenol A including chromatography, fluorescence, etc., however there is no discussion around disadvantages of these analytical methods and why there is a continuous need to develop alternative detection platforms.

Answer: We have added discussion around disadvantages of these analytical methods to the manuscript.

The revised part now reads:

However, these methods suffer from some disadvantages such as high cost, time consuming running, unsuitability for on-site analysis and require qualified operators and complicated instrumentations. Therefore, it is worth developing an alternative detection method for determining the trace levels of BPA.

-Line 75 – Electroanalytical techniques were mainly presented in the context of voltammetric methods. It would be helpful to demonstrate that Bisphenol A have been also successfully detected using other electroanalytical approaches including amperometry, impedance, field effect transistors etc.

Answer: Since Bisphenol A is mostly measured using voltammetric and amperometricmethods, we have mentioned voltammetric and amperometricmethods in the manuscript. Furthermore, we added to the manuscript that electrochemical methods are divided into different categories such as conductometric, potentiometric, impedimetric, voltammetric, and amperometric techniques.

The revised part now reads:

With regard to the electrical signal, which should be measured, researchers classified the electrochemical procedures into conductometric, potentiometric, impedimetric, voltammetric, and amperometric techniques [9]. In recent time, the voltammetric and amperometric procedures play important roles in the detection of BPA.

Amperometry is one of a family of electrochemical methods in which the potential applied to a sensing electrode is controlled instrumentally and the current occurring as a consequence of oxidation/reduction at the electrode surface is recorded as the analytical signal. In its simplest form, the applied potential is stepped to and then held at a constant value; and resulting current is measured as a function of time. Voltammetric techniques employ measuring current at various potential points in a current- voltage curve in contrast to the fixed potential point in amperometric technique.

-Line 87 – While the authors describe challenges associated with the application of unmodified electrodes for electrochemical sensing, it would be helpful to put it in context of detecting Bisphenol A. Additional discussion on properties and redox mechanism of Bisphenol A should be included. Furthermore, it is unclear what is meant by bare electrode (e.g. graphene or graphite, glassy carbon, etc.) and why these electrodes suffer from reduced electron transfer kinetics? Is it due to adsorption of oxidation products of Bisphenol A on electrode surface during redox transformations? Discussion regarding surface characteristics and their influence on the electroanalytical performance of sensors would be helpful. It would be also helpful to include details on oxidation mechanism of Bisphenol A during electroanalytical measurements.

Answer: We added the oxidation mechanism of Bisphenol A to the manuscript and also explained why the surface of the electrode was contaminated during the measurement of bisphenol A.

The revised part now read:

Electrochemical detection of BPA is based on the well-known electroactivity of the phenolic groups present in the molecule (Figure 1). The electrooxidation process of BPA involved two electrons and two protons that the reaction mechanism is based on the oxidation of the phenolic groups from BPA structure (Figure 2) [11].

Figure 2. Electrooxidation mechanism of BPA.

Additionally, the oxidation products of BPA cause the fouling of the electrode surface which is the major problem occurs during the electrooxidation of phenols due to the electropolymerization of phenolic compounds [12]. The formation of polymeric product after BPA oxidation blocks the active surface area of the electrodes and delays important electrodes processes [13].

-Lines 93-95 - The authors state that for the surface modified electrodes “the compound enters the electrode matrix and alters its electro chemical features”. However, it is unclear what is meant by this statement. The detection mechanism will vary between different electrode surfaces (e.g. direct binding transduced electronically, surface interactions, etc.). It may be helpful for the reader to understand the most common interaction mechanisms between electrode surface material and Bisphenol A. In addition, discussion regarding electrode modifications needed to develop biosensors, where a biomaterial (e.g. enzyme, aptamer, antibody, etc.) which functions as recognition element is typically immobilised on the electrode surface, should be included.

Answer: Importantly, the recognition reaction between biomolecules and targets provides the biosensor with a high degree of selectivity for the target to be measured. Since selectivity is an important parameter in the design of electrochemical sensors, it is important to fabricate electrochemical sensors based on biological molecules.

-While the authors list numerous advantages of using modified electrodes for detecting analytes (lines 98 – 102), it would be helpful to include additional discussion on some disadvantages associated with extensive electrode surface modifications such as poor batch-to-batch reproducibility, often lengthy and complex preparation steps, etc.

Answer: We added some of these disadvantages as the challenge in the conclusion section.

The added part now reads:

Even though great achievements have been made for the past decades in this field, there still exist a lot of challenges for further investigations in BPA electrochemical sensors, including:

  1. The properties of nanomaterials depend on the size, shape, oxidation level, degree of purity, structural defects, and degree of dispersion, since these factors highly affect the electrical conductivity, catalytic activity, and physical chemical binding interactions and kinetics (between nanomaterials and other functional materials). In addition to, some synthetic routes of nanomaterials are tedious and special conditions required. Therefore, we need to search new synthesis method to produce uniform and stable nanomaterials and their composites for design of BPA sensors.
  2. The leakage or detachment of modifiers from sensor surface during the usage may produce profound influence on the overall performance of sensor. That is, the leakage or detachment of modifiers may decrease the activity sites on the sensing surface and reduce the electronic conductivity of sensor. Therefore, effective and robust immobilization methods for modifiers in BPA electrochemical sensors should be addressed carefully.
  3. Some electrode preparations are complex, time consuming and costly. Therefore, easy and low-cost modification of the electrode surface is very important to fabrication of BPA electrochemical sensors.
  4. The matrix effect when applying the sensors for real sample analysis. Samples like food and water always contain a mixture of ions, macromolecules, and other electroactive species that would induce a nonspecific signal in the BPA detection process. For this, one research direction is the biorecognition elements which should be specific to BPA and the other is the effective sample pretreatment methods. However, the efficient immobilization of recognizing elements for transducers remains a major challenge in biosensor technology.
  5. The commercialization of BPA electrochemical sensors is hampered due to some technical barriers. In many cases the main limitations in commercialization of electrochemical sensors are poor analytical performance in real samples, complex detection procedures, lack of a cost-effective production method. Therefore, further improvements in terms of miniaturization, portability, minimizing matrix effect, ability to conduct continuous measurements, validation parameters, and competitive cost are necessary.

Therefore, future research needs to resolve these challenges in order to improve the electrochemical determination of BPA.

-To improve clarity for the reader, it would be helpful to include a table summarising key electrochemical sensors developed for sensing Bisphenol A with a further breakdown to highlight sensor configuration, detection limits, electroanalytical techniques used, etc.

Answer: We added a table in manuscript for comparison of the performance of bisphenol A electrochemical sensors.

The added part now reads:

Table 1: Comparison of the performance of BPA electrochemical sensors and biosensors.

Electrochemical sensor

Method

Detection limit

Linear range

Ref.

c-MWCNTs/GCE

Voltammetry

5.0 nM

10-104 nM

17a

SWCNT–â-CD/GCE

Amperometry

1.0 nM

10.8 nM-18.5 µM

17b

MWCNTs-OH/GCE

Amperometry

0.81 µg L−1

1-24 µg L−1

17c

Gr/GCE

DPV

4.689×10-8 M

5×10-8-1×10-6 M

19

Fullerene/GCE

SWV

3.7 nM

74 nM−0.23 µM

21

4-MBA/gold nano-dendrites/GCE

DPV

1.2 nM

0.05-55.0 ìM

23

Fe3O4 NPs derivative/SPE

DPV

0.01 ìM

0.03-700.0 ìM

25

PEDOT/GCE

CV

55 ìM

90-410 ìM

27

Amperometry

22 ìM

40-410 ìM

CS/N-GrS/GCE

Amperometry

5.0×10−9 M

1.0×10−8-1.3×10−6 M

29a

AuNP/PVP/PGE

SWAdSV

1.0 nM

0.03–1.10 ìM

29b

MWCNT/PTh/Pt/GCE

DPV

3.0×10-9 M

5.0×10-8-4.0×10-7 M

29c

AuNP/MWCNT/GCE

DPV

4 nM

0.01-0.7 µM

29d

rGO-Ag/PLL/GCE

DPV

0.54 ìM

1-80 ìM

29e

AuNPs-rGO-MWCNTs/GCE

DPV

1.0×10−9 M

5.0×10−9−2.0×10−5 M

29f

SWCNTs/Poly-IL/GCE

DPV

1.0×10−9 M

5.0×10−9-3.0×10−5 M

31a

ZnO/CNTs/IL/CPE

SWV

9.0 nM

0.002-700 µM

31b

MIP/CS/Gr/ABPE

Derivative Voltammetry

6.0 nM

8.0−2.0 µM

33a

MIPPy/GQDs/GCE

DPV

0.04 ìM

0.1-50 ìM

33b

AuNPs/Gr/GCE

DPV

5 nM

0.01-10 µM

36a

f-MWCNTs/AuNPs/Au

SWV

0.05 nM

0.1-10 nM

36b

MWCNT/Fe3O4-SH@Au/

aptamer/MCH/GCE

DPV

0.03 nM

0.1-8 nM

36c

Tyr-DAPPT-rGO/GCE

Amperometry

3.5×10-10 M

1.0×10-9-3.8×10-5 M

38a

Lac/Ag–ZnO/MWCNTs/CSPE

DPV

6.0 nM

0.5–2.99 ìM

38b

BCNPs/Tyr/Nafion/GCE

Amperometry

3.18 nM

0.02-10 µM

38c

BSA/Anti-BPA/AuNPs/MWCNTs/GCE

LSV

8.7×10-9 M

1.0×10-8-1.0×10-6 M

40

-The rationale behind the inclusion of described studies is unclear. Are these key fundamental or first demonstrations of applying a specific material for developing a Bisphenol A sensors? Are these selected based on the best performance metrics including sensitivity, selectivity or detection range, etc. (to date) based on published literature? The description on how this review is organised with associated rationale should be included.

Answer: In analytical chemistry, the validation of a method is an essential step for demonstrating that the results of following an analytical procedure will be close enough to the unknown true value for the content of the analyte under study. In fact, the performance of analytical method can be validated by assessing its figures of merit (sensitivity, selectivity, limit of detection, repeatability, reproducibility, etc). Therefore, these are key fundamental in design and fabrication of electrochemical sensors.

-It would be helpful for the reader to include at the end of each section (e.g. CNT based electrochemical sensors, ionic liquids, etc.) a comprehensible discussion on future challenges associated with adaptation of these materials for making stable, sensitive and reproducible sensors. For example, changes in surface morphology of graphene, CNTs are known to strongly influence the response of studied molecules in electroanalytical measurements. In addition, challenges associated with high manufacturing costs, limited control over surface defects during synthesis, etc. in the context of making electroanalytical sensors for Bisphenol A should be discussed.

Answer: In general, we added challenges to the electrochemical detection of BPA in the conclusionssection.

The revised part now reads:

Even though great achievements have been made for the past decades in this field, there still exist a lot of challenges for further investigations in BPA electrochemical sensors, including:

  1. The properties of nanomaterials depend on the size, shape, oxidation level, degree of purity, structural defects, and degree of dispersion, since these factors highly affect the electrical conductivity, catalytic activity, and physical chemical binding interactions and kinetics (between nanomaterials and other functional materials). In addition to, some synthetic routes of nanomaterials are tedious and special conditions required. Therefore, we need to search new synthesis method to produce uniform and stable nanomaterials and their composites for design of BPA sensors.
  2. The leakage or detachment of modifiers from sensor surface during the usage may produce profound influence on the overall performance of sensor. That is, the leakage or detachment of modifiers may decrease the activity sites on the sensing surface and reduce the electronic conductivity of sensor. Therefore, effective and robust immobilization methods for modifiers in BPA electrochemical sensors should be addressed carefully.
  3. Some electrode preparations are complex, time consuming and costly. Therefore, easy and low-cost modification of the electrode surface is very important to fabrication of BPA electrochemical sensors.
  4. The matrix effect when applying the sensors for real sample analysis. Samples like food and water always contain a mixture of ions, macromolecules, and other electroactive species that would induce a nonspecific signal in the BPA detection process. For this, one research direction is the biorecognition elements which should be specific to BPA and the other is the effective sample pretreatment methods. However, the efficient immobilization of recognizing elements for transducers remains a major challenge in biosensor technology.
  5. The commercialization of BPA electrochemical sensors is hampered due to some technical barriers. In many cases the main limitations in commercialization of electrochemical sensors are poor analytical performance in real samples, complex detection procedures, lack of a cost-effective production method. Therefore, further improvements in terms of miniaturization, portability, minimizing matrix effect, ability to conduct continuous measurements, validation parameters, and competitive cost are necessary.

Therefore, future research needs to resolve these challenges in order to improve the electrochemical determination of BPA.

-Discussion regarding current technological challenges associated with the development and commercialisation of Bisphenol A sensors should be added to the “Conclusions” section. Discussion on the solutions that can be implemented to overcome these issues should be also included. Also, it would be helpful to provide a future outlook regarding which materials or sensor configurations demonstrate the most promising characteristics for developing stable, reproducible, sensitive and selective Bisphenol A sensors.

Answer: We added this in the conclusions section.

The revised part in the conclusion now reads:

Even though great achievements have been made for the past decades in this field, there still exist a lot of challenges for further investigations in BPA electrochemical sensors, including:

  1. The properties of nanomaterials depend on the size, shape, oxidation level, degree of purity, structural defects, and degree of dispersion, since these factors highly affect the electrical conductivity, catalytic activity, and physical chemical binding interactions and kinetics (between nanomaterials and other functional materials). In addition to, some synthetic routes of nanomaterials are tedious and special conditions required. Therefore, we need to search new synthesis method to produce uniform and stable nanomaterials and their composites for design of BPA sensors.
  2. The leakage or detachment of modifiers from sensor surface during the usage may produce profound influence on the overall performance of sensor. That is, the leakage or detachment of modifiers may decrease the activity sites on the sensing surface and reduce the electronic conductivity of sensor. Therefore, effective and robust immobilization methods for modifiers in BPA electrochemical sensors should be addressed carefully.
  3. Some electrode preparations are complex, time consuming and costly. Therefore, easy and low-cost modification of the electrode surface is very important to fabrication of BPA electrochemical sensors.
  4. The matrix effect when applying the sensors for real sample analysis. Samples like food and water always contain a mixture of ions, macromolecules, and other electroactive species that would induce a nonspecific signal in the BPA detection process. For this, one research direction is the biorecognition elements which should be specific to BPA and the other is the effective sample pretreatment methods. However, the efficient immobilization of recognizing elements for transducers remains a major challenge in biosensor technology.
  5. The commercialization of BPA electrochemical sensors is hampered due to some technical barriers. In many cases the main limitations in commercialization of electrochemical sensors are poor analytical performance in real samples, complex detection procedures, lack of a cost-effective production method. Therefore, further improvements in terms of miniaturization, portability, minimizing matrix effect, ability to conduct continuous measurements, validation parameters, and competitive cost are necessary.

Therefore, future research needs to resolve these challenges in order to improve the electrochemical determination of BPA.

-The English language throughout the manuscript must be improved.

Answer: We polished the English to have higher readability.

Reviewer 3 Report

The manuscript summarized the reported electrochemical sensors for Bisphenol A detection. The modification of the working electrode is emphasized. The manuscript needs more discussion of the reported references, not just list all of them.

  1. The author should prepare a table to compare the properties of the electrochemical sensors for Bisphenol A detection.
  2. The author should point out the challenges in this field.
  3. In line 213, the “AuCl4-” is not typed correctly.
  4. Can the reported electrochemical sensors meet the detection requirements of BPA in the environmental samples?
  5. Can the author explain the redox reaction and its CV or DPV plot of BPA on the nanomaterials modified electrode?
  6. The author should discuss and present more details of nanomaterials design and fabrication.
  7. Can the author compare the electrochemical sensors and electro-biosensors for the detection of BPA?

Author Response

-The manuscript summarized the reported electrochemical sensors for Bisphenol A detection. The modification of the working electrode is emphasized. The manuscript needs more discussion of the reported references, not just list all of them.

Answer: In general, we have explained the properties of each category of modifiers at the beginning of each section.

-The author should prepare a table to compare the properties of the electrochemical sensors for Bisphenol A detection.

Answer: We have added a comparison table for detection of BPA in the manuscript.

-The author should point out the challenges in this field.

Answer: We stated the challenges in this field in the manuscript.

Even though great achievements have been made for the past decades in this field, there still exist a lot of challenges for further investigations in BPA electrochemical sensors, including:

  1. The properties of nanomaterials depend on the size, shape, oxidation level, degree of purity, structural defects, and degree of dispersion, since these factors highly affect the electrical conductivity, catalytic activity, and physical chemical binding interactions and kinetics (between nanomaterials and other functional materials). In addition to, some synthetic routes of nanomaterials are tedious and special conditions required. Therefore, we need to search new synthesis method to produce uniform and stable nanomaterials and their composites for design of BPA sensors.
  2. The leakage or detachment of modifiers from sensor surface during the usage may produce profound influence on the overall performance of sensor. That is, the leakage or detachment of modifiers may decrease the activity sites on the sensing surface and reduce the electronic conductivity of sensor. Therefore, effective and robust immobilization methods for modifiers in BPA electrochemical sensors should be addressed carefully.
  3. Some electrode preparations are complex, time consuming and costly. Therefore, easy and low-cost modification of the electrode surface is very important to fabrication of BPA electrochemical sensors.
  4. The matrix effect when applying the sensors for real sample analysis. Samples like food and water always contain a mixture of ions, macromolecules, and other electroactive species that would induce a nonspecific signal in the BPA detection process. For this, one research direction is the biorecognition elements which should be specific to BPA and the other is the effective sample pretreatment methods. However, the efficient immobilization of recognizing elements for transducers remains a major challenge in biosensor technology.
  5. The commercialization of BPA electrochemical sensors is hampered due to some technical barriers. In many cases the main limitations in commercialization of electrochemical sensors are poor analytical performance in real samples, complex detection procedures, lack of a cost-effective production method. Therefore, further improvements in terms of miniaturization, portability, minimizing matrix effect, ability to conduct continuous measurements, validation parameters, and competitive cost are necessary.

Therefore, future research needs to resolve these challenges in order to improve the electrochemical determination of BPA.

-In line 213, the “AuCl4-” is not typed correctly.

Answer: We corrected it.

-Can the reported electrochemical sensors meet the detection requirements of BPA in the environmental samples?

Answer: According to the reported articles, the fabricated electrochemical sensors shows satisfactory recoveries for detection of BPA in real samples such as water samples, mineral water plastic bottle, polycarbonate water bottle, different brands of baby bottels, canned foods, milk, orange juice, etc. However, the presense of ions and other electroactivespesies in environmental samples can interfere with the detection of BPA in real samples.

-Can the author explain the redox reaction and its CV or DPV plot of BPA on the nanomaterials modified electrode?

Answer: We have explained the mechanism related to the process of oxidation BPA in the introduction section.

But in connection with the CV and DPV plots:

CV is the most important and widely used electroanalytical technique that gives first hand information about the oxidation process of analyte such as redox potential, electrochemical reaction rates,reversible or irreversible oxidation process and also diffusion or surface adsorption of process. Though CV is highly informative, its high back ground current imposes some limitations for its use in the determination of various molecules. Therefore more sensitive technique like DPV and SWV can be employed for the low level detection of a wide range of analytes. In fact, for quantitative measurements (determining the sensitivity and detection limit of the sensor, etc), techniques such as DPV, SWV with higher sensitivity are used.

-The author should discuss and present more details of nanomaterials design and fabrication.

-Can the author compare the electrochemical sensors and electro-biosensors for the detection of BPA?

Answer: According to the results of the various electrochemical sensors and biosensors for BPA detection, which have been reviewed, most of electrochemical biosensors show lower detection limits and higher sensitivities than electrochemical sensors. In fact, the affinity of the biorecognition elements (aptamers, enzymes and antibodies) toward the BPA provides specific detection and also increases the sensitivity and selectivity of the detection.

Round 2

Reviewer 2 Report

While the reviewers address all the comments, the English language throughout the manuscript should be improved prior to the publication.

Author Response

While the reviewers address all the comments, the English language throughout the manuscript should be improved prior to the publication.

Answer: We revised the manuscript to improve the English quality. The polished English is provided in the revised manuscript.

Reviewer 3 Report

The revised manuscript has a great improvement in both language and perspective.

Author Response

Answer: We further revised and polished the English of the manuscript.